# Full Solutions to Flow and Heat Transfer from Slip-Induced Microtube Shapes

**DOI:** 10.3390/mi14040894

**Published:** 2023-04-21

**Authors:** Mustafa Turkyilmazoglu, Faisal Z. Duraihem

**Affiliations:** 1Department of Mathematics, Hacettepe University, Beytepe, Ankara 06532, Turkey; 2Department of Medical Research, China Medical University Hospital, China Medical University, Taichung 404327, Taiwan; 3Department of Mathematics, College of Science, King Saud University, Riyad 11451, Saudi Arabia

**Keywords:** slip mechanism, new tubular microchannels, elliptical pipes, hydrodynamically developed, thermally developed, heat transfer

## Abstract

The main concern of this paper is to introduce some new tubular shapes whose cross-sections result from the imposition of Navier’s velocity slip at the surface. A new family of pipes induced by the slip mechanism is thus discovered. The family is shown to modify the traditional pipes with elliptical cross-sections in the absence of slip, and they partly resemble collapsible tubes. The velocity field through the new pipes is then analytically determined. Afterwards, the corresponding temperature field with a constant heat flux boundary is shown to be perturbed around the slip parameter, whose leading order is well known from the literature. The correction to this order is next evaluated analytically. The velocity and temperature fields are further discussed regarding such new shapes. More physical features, such as the wall shear stress, the centerline velocity, the slip velocity and the convective heat transfer are also studied in detail. From the solutions, it is observed that a circular pipe under the effect of a slip mechanism has the largest temperature and the lowest Nusselt number at the center of the modified pipe. The new pipes are thought to have engineering and practical value in the micromachining industry, besides offering new analytical solutions for the considered flow geometry.

## 1. Introduction

Fluid flowing through a circular or non-circular long pipe due to pressure variations has practical applications in various real life situations [1]. Engineers are interested in medical nanotube manufacturing, biomechanical microfluidic devices, microelectronic devices, and natural gas transportation to detect fully developed laminar fluid properties, as well as the thermally developed temperature distribution pertinent to the flowing medium, in order to assess pressure drops and pumping rates [2,3,4,5]. The present analysis focuses on new types of non-circular microchannels induced by Navier’s slip factor at the wall surface, taking into account the hydrodynamically and thermally developed flow and thermal conditions through analytical means.

Fully developed flow within special shapes of solid mechanics with different cross-sections has been assessed by complex analysis methods in an earlier publication [6]. In historical order, circular or non-circular pipes and laminar flow and heat transfer solutions were reported in [7]. The Graetz problem in the entrance region of irregular ducts was analyzed in [8]. With regard to pipes with elliptical cross-sections, numerical investigations were undertaken to assess flow and thermal fields in [9]. An analytical slip flow solution based on the Knudsen number was offered in [10]. The effects of velocity slip through elliptic cross-sections were studied numerically in [11]. A variety of cross-sectional pipes were generated together with the corresponding analytical velocity solutions in [12] by means of complex analytical functions. Through a scaling approach, pressure drop formulas were presented in [13] for flow in non-circular ducts relevant to heat exchanger applications. Tubes with varying geometries and the computation of the pressure losses in fluids of Newtonian and non-Newtonian type were considered in [14]. Straight ducts [15] or equilateral triangular ducts [16] were analytically explored with the aim of capturing the heat transfer characteristics. For partially collapsible tubes, one can refer to [17]. The authors of [18] analyzed gaseous slip flow behavior in symmetric and non-symmetric parabolic microchannels.

The literature shows that circular cylinders, as well as those with a rectangular cross-section, are the intensive focus of past and current research. For instance, heat transfer in thermally developing flow with a piecewise wall temperature within a concentric annulus was analytically analyzed for laminar and turbulent flow in [19]. The same problem for a multicomponent fluid with Soret and Dufour effects was examined in [20] with a prescribed heat flux in a circular pipe. A second-order slip model was constructed in [21] to determine the effects of a low Peclet number in microtube flow. The steady laminar flow of Newtonian fluids in tubes of arbitrary cross-sections was considered in [22]. Non-linear viscoelastic fluids in straight tubes of arbitrary cross-sections were accounted for in [23]. Ellis non-Newtonian fluids inside circular tubes were recently documented in [24]. It was shown that the Nusselt numbers for a shear-thinning Ellis fluid are higher than their counterparts in a Newtonian fluid. Tubes of arbitrary varying cross-sections have also been the topic of several theoretical and numerical studies in recent research works, see, for instance, [25,26,27,28,29,30,31]. Recently, non-linear vibrations of a hyper-elastic tube were examined with different physical models to estimate the frequency of the non-linear vibrations [32]. For different techniques of solving fluid flow and heat transfer and the potential applications of microchannels involving hydrophobic/hydrophilic walls, the reader can refer to articles [33,34].

The motivation behind the current research is to develop new tubular pipes whose cross-sections are induced by Navier’s velocity slip factor at the surface. The newly constructed pipes are initially shown to be modified traditional elliptical pipes; they can be considered as special collapsible tubes generated from the elliptical ones. Such collapsible tubes may certainly be encountered in human/animal eye models [35,36]. Indeed, in miniaturized microprocessor chips, a high heat flux should be handled by designing proper passages for coolant flows, so an efficient heat transfer is achieved by providing such pipes in computer microchannels with hydrophobic/slippery surfaces, please refer to [34] for more details. It is hence significant to emphasize that the slip rate of a selected cross-section will be assured over such slip-induced pipes. Laminar Newtonian viscous flows across such tubes and the resulting thermally developed temperature field are then analytically derived. The geometry and physical properties of these pipes are eventually studied in detail. Since laminar flow and heat transfer in elliptical tubes are known to be significant in microfluidic applications and in various heat exchangers, the presented tubes are presumed also to take part in such physical applications.

## 2. Equations of Momentum and Energy

Let us consider a symmetric tubular boundary as shown in Figure 1. The major axis is at $\bar{x}=\bar{a}$ and the minor axis is at $\bar{y}=\bar{b}$, both of which constitute the symmetry axes. The fluid flowing through the pipe, encountering a temperature distribution *T* across the domain, is thought to have only axial velocity $\bar{u}\left(r,\theta \right)$, where (r,θ) is the cylindrical coordinate system linked to the usual Cartesian coordinate system via
(1)$$\begin{array}{ccc} \bar{x} & = & \bar{a}r\cos\theta ,\\ \bar{y} & = & \bar{a}r\sin\theta . \end{array}$$

To make the coordinates dimensionless with a hydraulic diameter *D*, (Equation 1) is replaced with the dimensionless counterpart
(2)$$\begin{array}{ccc} x & = & ar\cos\theta ,\\ y & = & ar\sin\theta , \end{array}$$
where $\left(x,y\right)=\left(\frac{\bar{x}}{D},\frac{\bar{y}}{D}\right)$ and $\left(a,b\right)=\left(\frac{\bar{a}}{D},\frac{\bar{b}}{D}\right)$. Furthermore, making use of the dimensionless velocity *u* and temperature Θ by means of
(3)$$\begin{array}{ccc} u & = & \frac{\bar{u}}{u_{b}},\\ \Theta & = & \frac{T-T_{w}}{qD/k}, \end{array}$$
where $u_{b}$ is the bulk velocity, $T_{w}$ is the wall temperature, *q* is the constant heat flux at the wall and *k* is the thermal conductivity, it is possible to reduce the governing equations of the hydrodynamically and thermally fully developed flow motion and heat transfer into the set
(4)$$\begin{array}{ccc} u_{rr}+\frac{1}{r}u_{r}+\frac{1}{r^{2}}u_{\theta \theta } & = & -4\left(1+\frac{1}{\rho^{2}}\right),\\ \Theta_{rr}+\frac{1}{r}\Theta_{r}+\frac{1}{r^{2}}\Theta_{\theta \theta } & = & 4u(r,\theta ). \end{array}$$
where ρ=b/a is the aspect ratio. The right hand side of hydrodynamic flow in (Equation 4) is owing to the constant pressure gradient, which was absorbed into the radial flow as a result of normalization together with the factor $\frac{a^{2}b^{2}}{a^{2}+b^{2}}$. In addition, the right hand side of the thermal equation is due to the conductive heat transfer effect. We should remark that due to the symmetry of the flow and temperature as well as the Navier’s velocity slip assumption at the wall, the system in (Equation 4) is complemented with the boundary constraints:(5)$$\begin{array}{ccc} u_{r} & = & 0,\;\Theta_{r}=0\;at\;r=0,\\ u & = & -Lu_{r},\;\Theta =0\;at\;r=R\left(\rho ,L,\theta \right), \end{array}$$
where *L* is the constant slip parameter and R(ρ,L,θ) is the shape function of the boundary to be determined later. We note that the slip condition in (Equation 5) is imposed strictly in the *r*-direction defined in (Equation 1), and by the help of temperature definition in (Equation 3), the surface temperature boundary condition in (Equation 5) follows.

The hydraulic diameter, bulk velocity and bulk temperature are further defined as [37]:(6)$$\begin{array}{ccc} D & = & \frac{4A}{P},\\ u_{b} & = & \frac{1}{A}\int_{A}\bar{u}dA,\\ T_{b} & = & \frac{1}{Au_{b}}\int_{A}\bar{u}TdA, \end{array}$$
where *A* is the area representing a cross-section of the modified pipe and *P* is the perimeter.

The rate of heat transferred from the pipe is measured by means of the Nusselt number
(7)$$\begin{array}{ccc} Nu & = & -\frac{qD}{k(T_{w}-T_{b})}. \end{array}$$

Hence, the dimensionless Nusselt number is, with the above considerations, given by
(8)$$\begin{array}{ccc} Nu & = & -\frac{1}{\Theta_{b}}\;=\;-\frac{A}{\int_{A}u\Theta \;dA}. \end{array}$$

## 3. Full and Perturbation Solutions

### 3.1. Full Velocity Solution and Pipe Surface Formation

Initially, we wish to determine the flow field solution of Equation (Equation 4) inside a suitably modified tubular shape which is symmetric about the *x*- and *y*-axes, as revealed in Figure 1. The shape function r=R is a result of the simultaneous solution of (Equation 4) together with the flow boundary conditions in Equation (Equation 5). Thus, our approach is quite distinct from the classical wall slip imposition in [10,11]. Choosing such a slip profile will guarantee the velocity slip rate occurrence.

We should note that the flow solution of (Equation 4) satisfies the following homogenous (the first two terms) and non-homogenous parts (the last term):(9)$$\begin{array}{ccc} u(r,\theta ) & = & -\left(1+\frac{1}{\rho^{2}}\right)r^{2}+\alpha +\beta r^{2}\cos\left(2\theta \right), \end{array}$$
where α and β are constants to be found from the boundary conditions in (Equation 5). On the one hand, since the slip constraints in (Equation 4) together with (Equation 2) clearly imply that
(10)$$\begin{array}{ccc} u(r=1,\theta =0) & = & -Lu_{r}\left(r=1,\theta =0\right),\\ u(r=\rho ,\theta =\pi /2) & = & -Lu_{r}\left(r=\rho ,\theta =\pi /2\right), \end{array}$$
we can determine α and β from (Equation 10) in the form
(11)$$\begin{array}{ccc} \alpha & = & \frac{2\left(1+2L\right)\left(2L+\rho \right)\left(1+\rho^{2}\right)}{\rho \left(1+\rho^{2}+2L\left(1+\rho \right)\right)},\\ \beta & = & -\frac{\left(-1+\rho \right)\left(1+2L+\rho \right)\left(1+\rho^{2}\right)}{\rho^{2}\left(1+\rho^{2}+2L\left(1+\rho \right)\right)}. \end{array}$$

On the other hand, since the slip constraint in (Equation 5) should be satisfied at all polar angles θ, the shape of the pipe is determined to be
(12)$$\begin{array}{ccc} R & = & -L+\sqrt{L^{2}+\frac{2(1+2L)\rho (2L+\rho )}{1+\rho^{2}+2L\left(1+\rho \right)+\left(-1+\rho \right)\left(1+2L+\rho \right)\cos\left(2\theta \right)}}. \end{array}$$

It is noted that the boundary of the pipe found in (Equation 12) depends not only the ratio ρ and polar angle θ, but also depends on the slip parameter *L* as dictated in (Equation 5). Therefore, it is named hereafter as the “slip-induced pipe”.

### 3.2. Perturbative Temperature Solution

The closed form solution to the energy Equations (Equation 4) and (Equation 5) with non-zero values of *L* is not accessible. Therefore, we plan to obtain a perturbation solution around L=0 to determine the influence of the slip mechanism (at least at small magnitudes). To achieve this, we apply the subsequent expansion
(13)$$\begin{array}{ccc} \Theta & = & \left(a_{1}+La_{2}\right)r^{4}+\left(b_{1}+Lb_{2}\right)r^{2}+\left(c_{1}+Lc_{2}\right)r^{4}\cos\left(2\theta \right)+\left(d_{1}+Ld_{2}\right)\\ & + & \left(e_{1}+Le_{2}\right)r^{2}\cos\left(2\theta \right)+\left(f_{1}+Lf_{2}\right)r^{4}\cos\left(4\theta \right)+O\left(L^{2}\right). \end{array}$$

By substituting (Equation 15) into (Equation 4), together with the consideration of temperature boundary conditions in (Equation 5), the leading and first order coefficients appear to satisfy the relations
(14)$$\begin{array}{ccc} a_{1} & = & \frac{1+\rho^{2}}{4\rho^{2}},\;a_{2}=0,\;b_{1}=-1,\;b_{2}=-\frac{4\left(1+\rho^{3}\right)}{\rho +\rho^{3}},\\ c_{1} & = & \frac{-1+\rho^{2}}{3\rho^{2}},\;c_{2}=-\frac{4(-1+\rho )}{3\rho \left(1+\rho^{2}\right)},\;d_{1}=\frac{2\left(5\rho^{2}+26\rho^{4}+5\rho^{6}\right)}{3\left(1+\rho^{2}\right)\left(1+6\rho^{2}+\rho^{4}\right)},\\ d_{2} & = & \frac{8\left(5\rho +54\rho^{3}+7\rho^{4}+164\rho^{5}+58\rho^{6}+58\rho^{7}+164\rho^{8}+7\rho^{9}+54\rho^{10}+5\rho^{12}\right)}{3{\left(1+7\rho^{2}+7\rho^{4}+\rho^{6}\right)}^{2}},\\ e_{1} & = & -\frac{2\left(-3-11\rho^{2}+11\rho^{4}+3\rho^{6}\right)}{3\left(1+7\rho^{2}+7\rho^{4}+\rho^{6}\right)},\\ e_{2} & = & -\frac{4\left(-3-19\rho^{2}-13\rho^{3}+6\rho^{4}-93\rho^{5}+230\rho^{6}-230\rho^{7}+93\rho^{8}-6\rho^{9}+13\rho^{10}+19\rho^{11}+3\rho^{13}\right)}{3\rho {\left(1+7\rho^{2}+7\rho^{4}+\rho^{6}\right)}^{2}},\\ f_{1} & = & \frac{\left(1+\rho^{2}\right)\left(1-2\rho^{2}+\rho^{4}\right)}{12\rho^{2}\left(1+6\rho^{2}+\rho^{4}\right)},\;f_{2}=-\frac{4\left(1-\rho +\rho^{2}-\rho^{3}-\rho^{4}+\rho^{5}-\rho^{6}+\rho^{7}\right)}{3\rho {\left(1+6\rho^{2}+\rho^{4}\right)}^{2}}. \end{array}$$

## 4. Results and Discussions

The newly discovered slip-induced tubular shapes and their flow/thermal properties will be discussed in this section.

It is easy to initially realize that when *L* is set to zero, the boundary of pipes from Equation (Equation 12) turns out to be
(15)$$\begin{array}{ccc} R(\rho ,\theta ) & = & \sqrt{2}\rho \sqrt{\frac{1}{1+\rho^{2}+\left(-1+\rho^{2}\right)\cos\left(2\theta \right)}}, \end{array}$$
which are the well-known traditional elliptical cross-sectional pipes, some of whose sketches are displayed in Figure 2 for a variety of aspect ratios. Thus, we are confident with the above mathematical analysis.

Figure 3, Figure 4, Figure 5 and Figure 6 are next shown to reveal the effects of slip parameter *L* with aspect ratios less than unity on the formation of various pipe boundaries.

It is apparent that the glass-like shaped tubes for small aspect ratios evolve into a more circular structure as ρ approaches 1, and the slip parameter seems to no longer effect the shape formation, since ρ=1 limits the classical circular pipe. Then, the slip factor only affects the flowing fluid inside. Further profiles corresponding to several slip factors with ρ>>1 are demonstrated in Figure 7a,b.

It can be shown that at the limit of large slip parameter, where L→∞, the limiting pipe shapes are governed by the below formula, obtained from (Equation 12):(16)$$\begin{array}{ccc} R & = & \frac{2\rho }{1+\rho +(-1+\rho )\cos(2\theta )}, \end{array}$$
the pictures of which are exhibited in Figure 8. It is also noted that the essential advantage of slip-induced pipes, as demonstrated in Figure 3, Figure 4, Figure 5, Figure 6, Figure 7 and Figure 8, is that once the physical slip rate is known, as can be experimentally or empirically measured, the slip at that rate will be physically guaranteed over such profiles.

In the case of L=0, a tube of elliptical cross-section as obtained from (Equation 15) possesses a well-documented cross-sectional area *A*, perimeter *P*, volumetric flow rate *Q* and Poiseuille number Pos, which is the product of friction factor and Reynolds number, all given by the following relations [38]:(17)$$\begin{array}{ccc} A & = & 2\int_{0}^{\frac{\pi }{2}}R^{2}d\theta =\pi \rho ,\\ P & = & 4\int_{0}^{\frac{\pi }{2}}\sqrt{R^{2}+{R\prime }^{2}}d\theta =4E,\\ Q & = & \int_{0}^{\frac{\pi }{2}}\int_{0}^{R}\frac{\rho^{2}}{1+\rho^{2}}ru\left(r,\theta \right)drd\theta =\frac{\pi \rho^{3}}{4(1+\rho^{2})},\\ Pos & = & 8\frac{A^{3}}{QP^{2}}, \end{array}$$
in which $E=\int_{0}^{\frac{\pi }{2}}\sqrt{1-e^{2}\sin^{2}\theta }d\theta$, which is the complete Elliptic integral with the eccentricity of the ellipse $e^{2}=1-\rho^{2}$. Note also that the pressure drop can be calculated as the inverse of *Q* in (Equation 17). Hence, the engineering goal of a reduction in the pressure drop can be achieved with non-circular and higher aspect ratio pipe structures. On the other hand, for such larger radius pipes, an enhanced power may be required to activate the fluid. Therefore, a balance should be sought in practical pumping conditions. Moreover, the Poiseuille number, Pos, expressed in terms of geometric parameters will be more useful for arbitrary shapes. For the slip-induced tubular shapes in (Equation 12), no analytical results can be found. However, some selected profiles and their physical characters are listed in Table 1, as numerically calculated from (Equation 17). The analytical results as displayed are fully compatible with the results in [7,11,13] in the case of traditional elliptical microchannels.

In parallel to the available data in [9], the friction factor and hence the Pos number decrease with an increase in the aspect ratio of the ellipse (up to the circular tube). As compared to the no-slip case, slip increases the volumetric flow rate steadily and reduces the friction factor. As a result, the Poiseuille number gradually decreases in the slip-induced pipes. However, as ρ increases further, the Poiseuille number also considerably increases with slip. With the limit of a large *L*, making use of (Equation 17), we are able to detect the following simple formula:(18)$$\begin{array}{ccc} A & = & \frac{1}{2}\pi \sqrt{\rho }\left(1+\rho \right), \end{array}$$
which is further consistent with the information provided in Table 1. The physics behind the contradictory effect of slip on the Poiseuille number in large and small aspect ratios is believed to be due to sharp changes in the velocity profiles and the endurance against the friction, which will be discussed next.

In terms of the Cartesian coordinates in (Equation 2), the main flow solution in (Equation 9) reads:(19)$$\begin{array}{ccc} u(x,y) & = & -\frac{2\left(1+2L\right)y^{2}\left(1+\rho^{2}\right)}{a^{2}\rho^{2}\left(1+\rho^{2}+2L\left(1+\rho \right)\right)}+\frac{2\left(1+2L\right)\left(2L+\rho \right)\left(1+\rho^{2}\right)}{\rho \left(1+\rho^{2}+2L\left(1+\rho \right)\right)}\\ & - & \frac{2x^{2}\left(2L+\rho \right)\left(1+\rho^{2}\right)}{a^{2}\rho \left(1+\rho^{2}+2L\left(1+\rho \right)\right)}, \end{array}$$
which turns out to be the classical pipe flow in elliptical cross-sections when L=0:(20)$$\begin{array}{ccc} u(x,y) & = & 2\left(1-\frac{x^{2}}{a^{2}}-\frac{y^{2}}{a^{2}\rho^{2}}\right). \end{array}$$

Some selected velocity profiles are demonstrated in Figure 9a,b, Figure 10a,b and Figure 11a,b to observe the impacts of ρ and *L*.

Note that in the case of ρ=1, we have the classical circular pipe solution
(21)$$\begin{array}{ccc} u(r,\theta ) & = & 2(1-r^{2}). \end{array}$$

The slip velocity becomes higher at the minor axis for ρ<1, but the maximum is attained at the major axis for ρ>1.

In addition to this, the centerline velocity can be computed from the relation
(22)$$\begin{array}{ccc} u(r=0) & = & \frac{2\left(1+2L\right)\left(2L+\rho \right)\left(1+\rho^{2}\right)}{\rho \left(1+\rho^{2}+2L\left(1+\rho \right)\right)}. \end{array}$$

Figure 12 exhibits some centerline velocity plots at selected slip parameters. The constant value of two of the centerline velocity is due to the Formulas (Equation 20) and (Equation 22), also in line with the conventional pipe solution in (Equation 21). The figure also shows that it is possible to find u(r=0) approaching infinity when ρ is very small and *L* is finite. This is apparent by taking the limit ρ→0 in Equation (Equation 22). Although it may be expected that the maximal velocity at the core is enlarged with the assistance of velocity slip in increasingly thinner pipes, its enormous limiting values may not be conceived reasonable since this is a Poiseulle flow subjected to a given pressure gradient.

The net wall shear stress exerted on the tube wall by the fluid, normalized based on pressure gradient and hydraulic diameter, is defined by means of
(23)$$\begin{array}{ccc} \tau & = & \sqrt{u_{r}^{2}+{\left(\frac{1}{r}u_{\theta }\right)}^{2}}{|}_{r=R}. \end{array}$$

Equation (Equation 23) can then be used to evaluate the drag force in micro-channels of cross-sections introduced here. The limiting values are found to be
(24)$$\begin{array}{ccc} \tau_{L=0} & = & 4\sqrt{1+\frac{1}{\rho^{2}}-\frac{2}{1+\rho^{2}+\left(-1+\rho^{2}\right)\cos\left(2\theta \right)}},\\ \tau_{L\to \infty } & = & \frac{4\sqrt{2}\left(1+\rho^{2}\right)\sqrt{1+\rho^{2}+\left(-1+\rho^{2}\right)\cos\left(2\theta \right)}}{\rho (1+\rho )(1+\rho +(-1+\rho )\cos(2\theta ))}. \end{array}$$

Figure 13a–d reveal the wall shear behaviors at different aspect ratios. In line with the physical intuition, the slip reduces the total wall shear to lead to a lower drag force, see [39]. In the case of the traditional elliptic pipe, the minimum shear remains on the major axis (θ=0) and the maximum shear on the minor axis (θ=π/2). For ρ>1, the above scenario is reversed. However, if stronger slip is applied, the minimum/maximum remains on the same axis, whereas the maximum/minimu wall shears attain their values somewhere between the minor and major axes, in particular for smaller and larger aspect ratios. As suggested from the velocity profiles in Figure 9, Figure 10 and Figure 11, profiles with a smaller aspect ratio are likely to produce higher velocity gradients at the azimuths, resulting in larger shears at the tube surface compared to those of bigger aspect ratio pipes. Moreover, the existence of a bimodal distribution of wall shear stress for smaller pipes can be attributed to the higher velocities prevalent in such shapes.

As for the thermal field concerning the slip-induced tubular pipes, the centerline temperature $\Theta_{c}=-\Theta \left(r=0\right)$ and the Nusselt number can be determined from the temperature solutions (Equation 13) and (Equation 14) in the forms
(25)$$\begin{array}{ccc} \Theta_{c} & = & \frac{\left(\frac{2\left(5\rho^{2}+26\rho^{4}+5\rho^{6}\right)}{3\left(1+\rho^{2}\right)\left(1+6\rho^{2}+\rho^{4}\right)}+\frac{8L\left(5\rho +54\rho^{3}+7\rho^{4}+164\rho^{5}+58\rho^{6}+58\rho^{7}+164\rho^{8}+7\rho^{9}+54\rho^{10}+5\rho^{12}\right)}{3{\left(1+7\rho^{2}+7\rho^{4}+\rho^{6}\right)}^{2}}\right)E^{2}}{\pi^{2}\rho^{2}}, \end{array}$$
(26)$$\begin{array}{ccc} Nu & = & \frac{9\pi^{2}\left(1+7\rho^{2}+7\rho^{4}+\rho^{6}\right)}{\left(17+98\rho^{2}+17\rho^{4}\right)E^{2}}\\ & - & L9\pi^{2}\left(1+\rho^{2}\right){\left(1+6\rho^{2}+\rho^{4}\right)}^{2}\left(3+\rho \left(-2+3\rho \right)\right)\\ & / & (119+\rho (-102+\rho (1511+\rho (-1184+\rho (5730+\rho (-3700+\rho (5730\\ & + & \rho \left(-1184+\rho \left(1511+17\rho \left(-6+7\rho \right)\right)\right))))))))E^{2}. \end{array}$$

Figure 14a,b are generated from (Equation 25) and (Equation 26). It is apparent that the centerline temperature is increased by the slip, which causes a lower heat transfer rates compared to the no slip flow case.

In line with the previous data in [9] for L=0, the Nusselt number is enhanced by the aspect ratios smaller from the one with the classical Nu value of 4.36 (or by an aspect ratio larger than unity). As shown in the circular cross-section, the Nusselt number and the centerline temperature are altered by the change in the aspect ratio and the slip parameter. It is interesting to anticipate that as the aspect ratio tends to zero, the lower limit of the Nusselt number of 5.2 is reached, the asymptotic value of which changes with the slip parameter as the aspect ratio becomes very large. However, slip always leads to a degradation in the heat transfer rate, because of the fact that slip yields a heated flow within the pipe. Finally, the reason we deliberately picked smaller values of *L* for temperature solutions is that they are produced from a small perturbation *L* analysis, valid only up to order of *L*, as can be inferred from pages 8 and 9. Higher values of *L*, therefore, are avoided not to present misleading behavior, otherwise they must be justified from a full numerical simulation.

## 5. Conclusions

Obtaining a new family of non-circular pipes as a result of the wall slip velocity is the main contribution of the current research. The modified tubular shapes are shown to transform into the classical elliptical microchannels when the slip is omitted. Therefore, the velocity, temperature and other physical characteristics of the new pipes exactly match available results in the open literature. The essential advantage of the present slip-induced pipes as considered here is that once the slip rate is known, as can be experimentally or empirically measured, the slip at that rate will be physically guaranteed over such profiles.

The following main conclusions can be reached from the analyses of present results:For small aspect ratios, the slip increases the volumetric flow rate continuously and it decreases the skin friction factor, leading to a reduction in the Poiseuille number.For large aspect ratios, the Poiseuille number increases with slip.For aspect ratios less than unity, the slip velocity becomes larger on the minor axis, with a maximum on the major axis for larger aspect ratios.The total shear stress helps the surface to achieve a lower drag force under the influence of the slip mechanism.In the presence of a stronger slip, the maximum/minimum shear is off the minor axis.The centerline temperature is reduced by the presence of slip, which results in lower heat transfer rates compared to no slip profiles.

Finally, the transition from laminar to turbulent flow inside the discovered microchannels can be explored in future studies.

## Figures and Tables

**Figure 1 micromachines-14-00894-f001:**
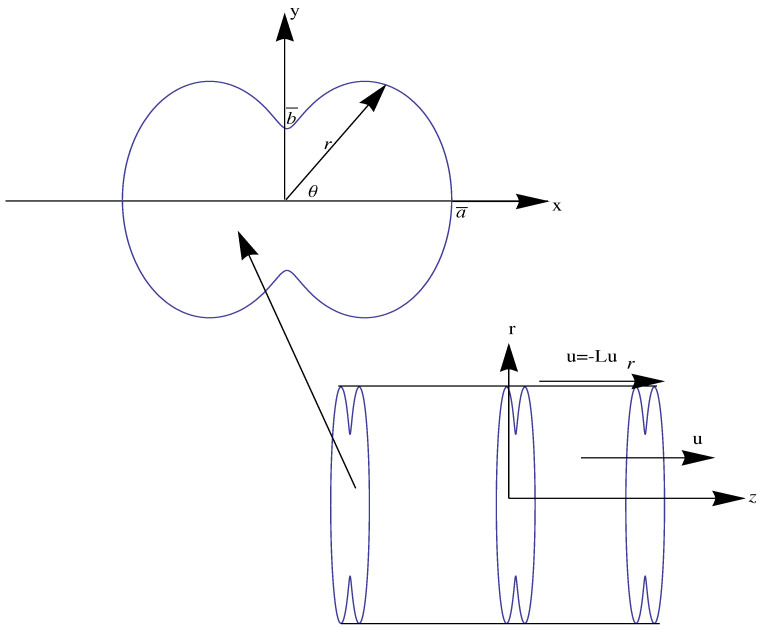
A symmetric pipe shape with Navier’s velocity slip character.

**Figure 2 micromachines-14-00894-f002:**
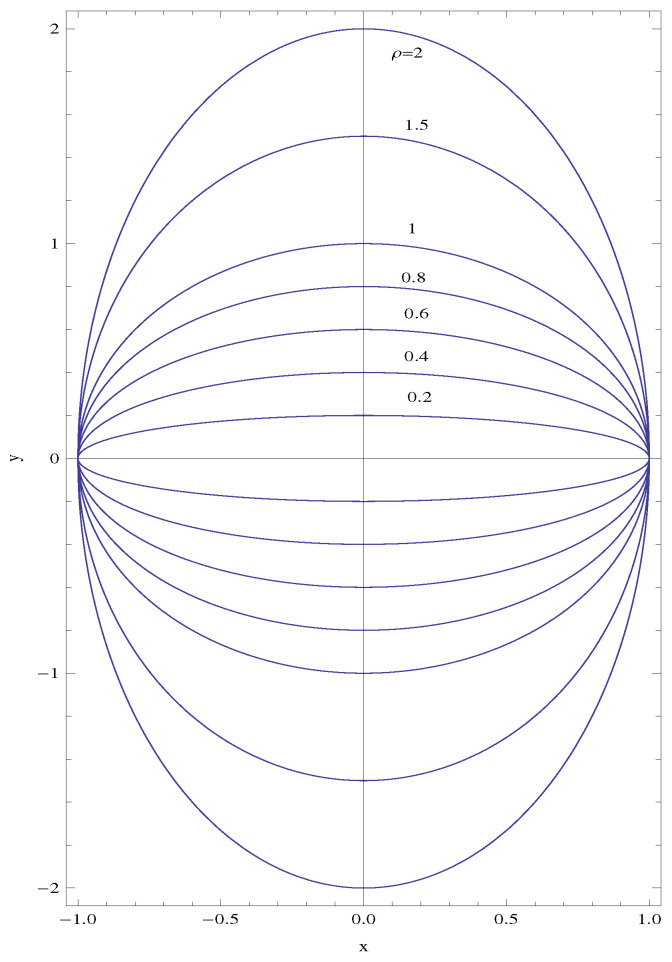
Elliptic pipe profiles corresponding to the no-slip flow boundary condition with L=0 in (Equation 15).

**Figure 3 micromachines-14-00894-f003:**
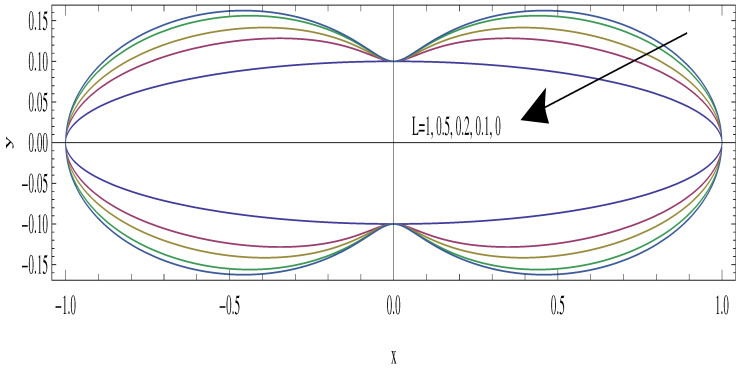
Slip-induced pipes with ρ=1/10.

**Figure 4 micromachines-14-00894-f004:**
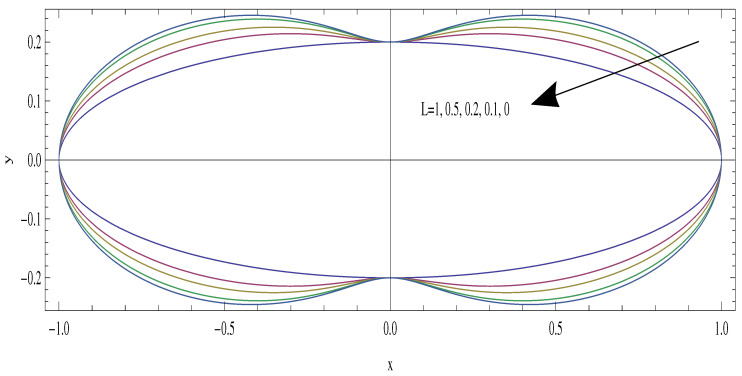
Slip-induced pipes with ρ=1/5.

**Figure 5 micromachines-14-00894-f005:**
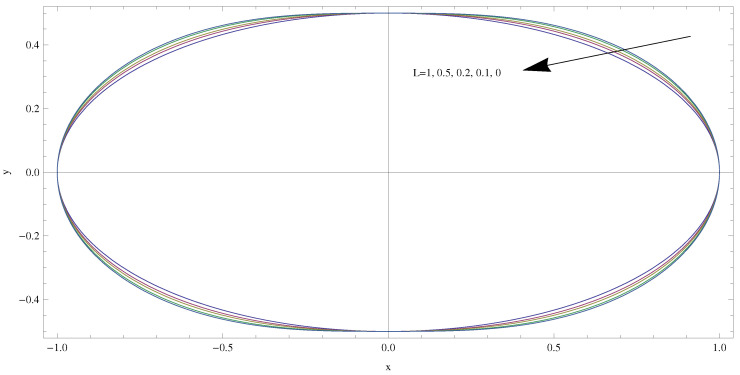
Slip-induced pipes with ρ=1/2.

**Figure 6 micromachines-14-00894-f006:**
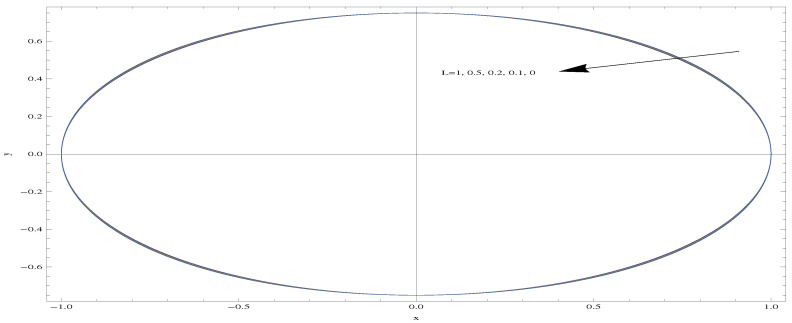
Slip-induced pipes with ρ=3/4.

**Figure 7 micromachines-14-00894-f007:**
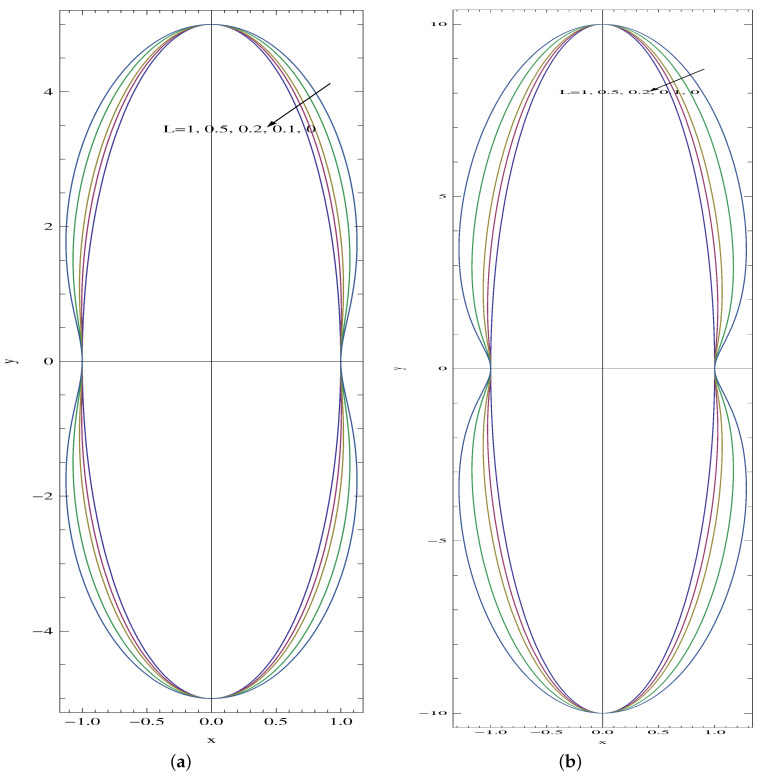
Slip-induced pipes with (**a**) ρ=5 and (**b**) ρ=10.

**Figure 8 micromachines-14-00894-f008:**
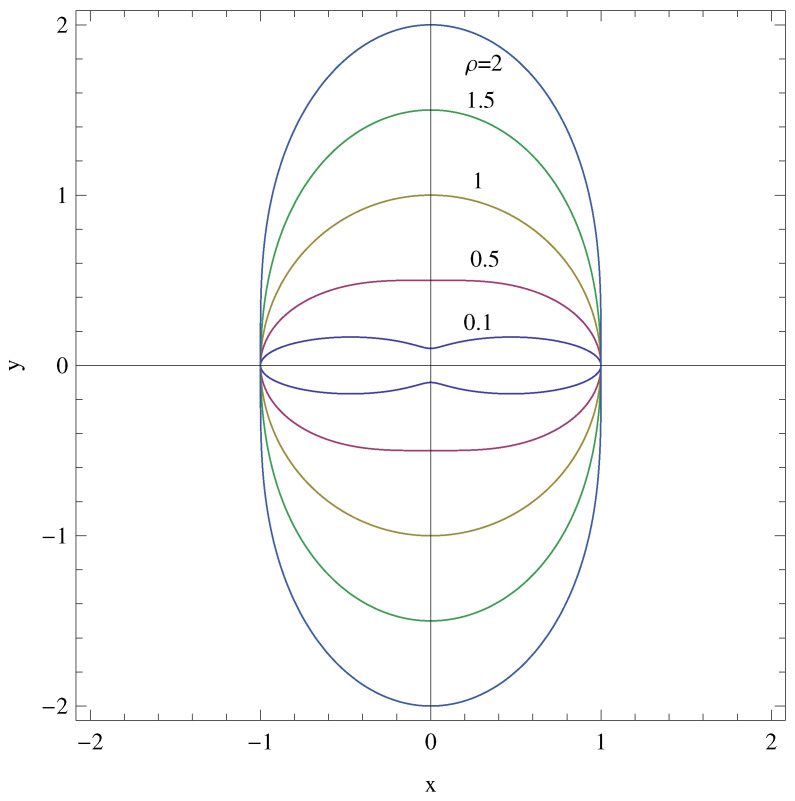
Slip-induced pipes with L→∞ at certain aspect ratios.

**Figure 9 micromachines-14-00894-f009:**
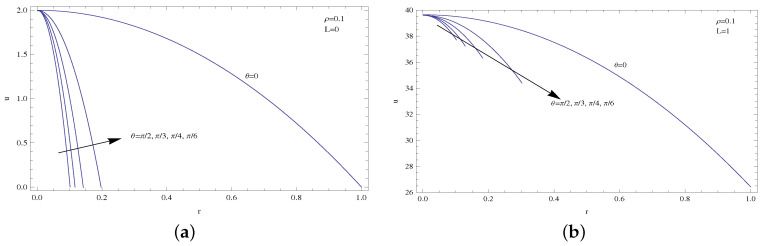
Velocity profiles for ρ=0.1. (**a**) L=0 and (**b**) L=1.

**Figure 10 micromachines-14-00894-f010:**
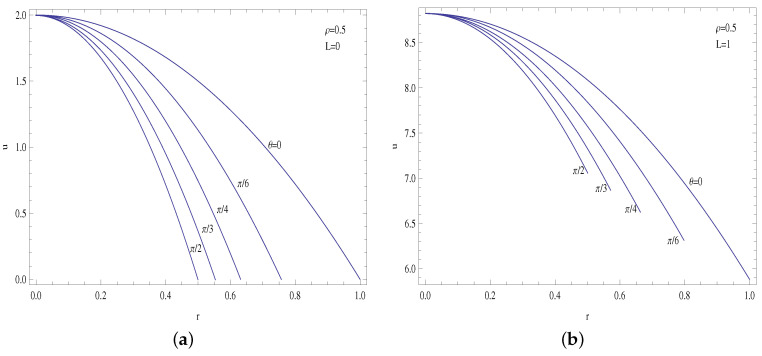
Velocity profiles for ρ=0.5. (**a**) L=0 and (**b**) L=1.

**Figure 11 micromachines-14-00894-f011:**
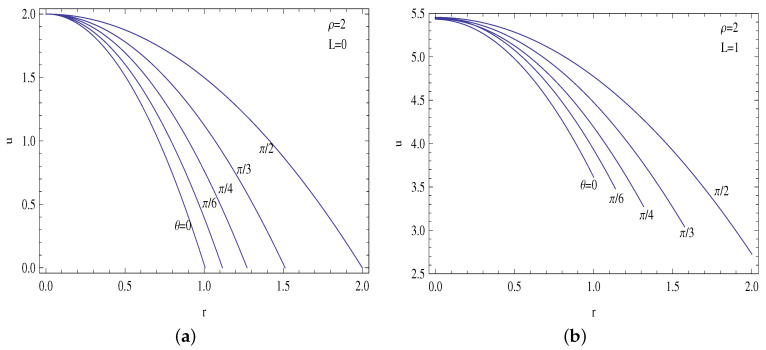
Velocity profiles for ρ=2. (**a**) L=0 and (**b**) L=1.

**Figure 12 micromachines-14-00894-f012:**
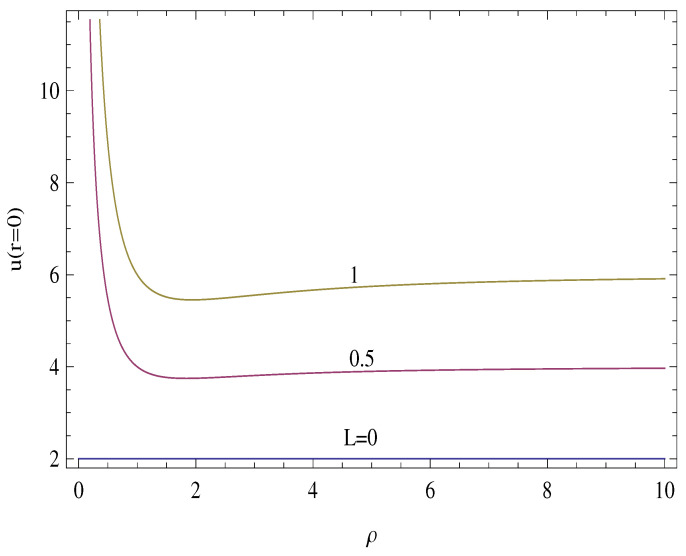
Centerline velocities for some values of *L* against different aspect ratios.

**Figure 13 micromachines-14-00894-f013:**
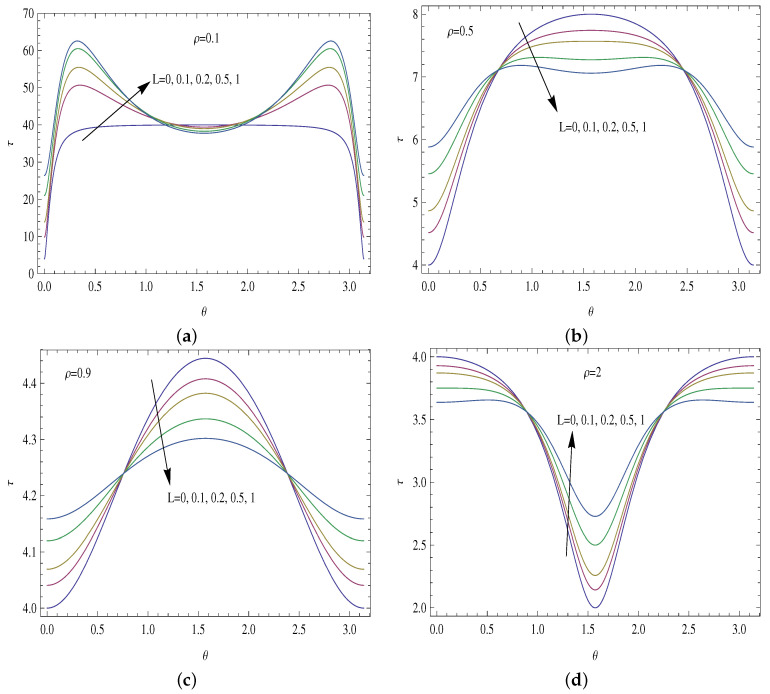
Wall shear profiles. (**a**) ρ=0.1, (**b**) ρ=0.5, (**c**) ρ=0.9 and (**d**) ρ=2 at chosen slip lengths.

**Figure 14 micromachines-14-00894-f014:**
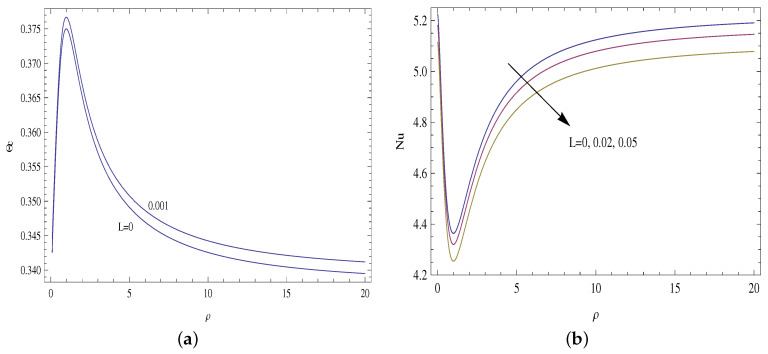
(**a**) Centerline temperature and (**b**) Nusselt number against different aspect ratios at selected slip lengths.

**Table 1 micromachines-14-00894-t001:** Geometrical and physical properties of slip-induced tubular shapes.

	*A*	*P*	*Q*	Pos
ρ=0.1 L=0	0.31416	4.06397	0.00078	19.3139
ρ=0.1 L=1	0.53260	4.23009	0.04548	1.48516
ρ=0.5 L=0	1.57080	4.84422	0.07854	16.8233
ρ=0.5 L=1	1.65528	4.94139	0.62797	2.36630
ρ=1.0 L=0	3.14159	6.28319	0.39270	16.0000
ρ=1.0 L=1	3.14159	6.28319	1.96350	3.20000
ρ=5.0 L=0	15.7080	21.0100	3.77595	18.6024
ρ=5.0 L=1	18.9085	21.5626	18.1058	6.42452
ρ=10 L=0	31.4159	40.6397	7.77622	19.3139
ρ=10 L=1	42.2733	41.2846	38.9915	9.09372

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
