# Peer review of "Full Solutions to Flow and Heat Transfer from Slip-Induced Microtube Shapes"

_micromachines, 2023, doi:10.3390/mi14040894_

Round 1
Reviewer 1 Report
The comments were attached.

Author Response
Dear Reviewer
We would like to thank you for providing a fair and encouraging
review of the paper. Please find below our response to your specific
points:
“1. The quality of writing of the manuscript should be increased.”
Thank you for the comment. The manuscript has been carefully
proofread and best effort has been spent to improve the writing quality.
“2. The definition of the Nusselt number in Eq. 7 is not correct.”
Thank you for the remark. It has been corrected now, please see
page 7 line 2.
“3. Different techniques for solving fluid flow and heat transfer and the
applications of microchannels should be discussed in the introduction. In this
regard, the following papers could be added to the introduction.”
Thank you for the remark. The provided references are found
relevant and much useful to improve the Introduction, and hence
they have been added now, please refer to page 3 lines 3-5 from the
bottom.
“4. The discussion of the results should be added to the manuscript. The
authors should describe the physics behind the results.”
Thank you for the remark. More physics have been inserted now
to discussion part, please refer to the highlighted parts throughout
the text.
“5. Authors claimed that the slip condition has a contradictory effect on
the Poiseuille number in large and small aspect ratios. Please describe this
phenomenon physically.”
Thank you for the remark. Some physics regarding the effect of
slip on the Poiseuille number in large and small aspect ratios have
been inserted now, please refer to page 14 last 2 lines and page 15
lines 1-2 after Table 1.
“6. The captions of figures do not have a clear description of them e.g., in
Fig. 14, it should be mentioned that the figures are for different aspect ratios.”
Thank you for the remark. The required corrections have been
fulfilled now, please refer to highlighted figure captions throughout
the text.
“7. In the introduction references should be cited as [2-5] not [2], [3], [4],
and [5]. These typos should be corrected for other references.”
Thank you for the remark. The required correction has been fulfilled
now, please refer to pages 2 and 3.
“8. In table 1, the units of parameters should be added.”
Thank you for the remark. Please note that parameters in Table
1 are unitless, please refer to Eq.(17) on page 12.
Reviewer 2 Report
The authors presented a full solution for the flow and heat transfer from slip induced microtube shapes. The authors should answer the following questions:
1. The authors mention that these shapes have engineering and practical value in micromachines. Can you provide some specific examples?
2. What are the assumptions of this study?
3. The definition of the dimensionless temperature doesn't seem dimensionally correct. You should double-check to make sure this parameter is dimensionless.
4. The convective terms of the momentum equation drop off due to one of the assumption that only axial velocity is non-zero. Therefore, only the effects of pressure gradient and viscous effect are considered in the study. However, I don't see how the pressure gradient in the axial direction can be normalized to become the right terms. You should provide some explanation showing the steps. I expect a dimensionless parameter to show up in the equation.
5. Once again, the convective terms of the energy equation drop off. Because of the lack of explanation in the manuscript, I don't understand why the conduction in the fluid is simply related to 4 times of the velocity component. What effect does the velocity component represent in the equation?
6. The authors claimed that the walls are subjected to constant heat flux. This contradicted to the boundary condition.
7. Based on conventional definition of Nusselt number, I am unable to derive equation (7). Do show your complete steps.
8. Theta_b was used to calculate Nusselt number, but how is Theta_b calculated?
9. What if L is greater than unity? Do provide some cases with L > 1.
10. Why does u(r=0) always equals to 2 when L=0 regardless of the values of rho? Please refer to Fig. 12 and explain.
11. Would it be reasonable to find that u(r=0) approaches infinity when rho is very small and L is not equal to zero? Please refer to Fig. 12 and explain. Do remember that this is a Poiseulle flow subjected to a given pressure gradient.
12. The definition of wall shear stress is not provided in the manuscript.
13. My intuition suggested me that something peculiar happens in Fig. 13(a). Do explain why there exits a bimodal distribution of wall shear stress. Also, explain why the average wall shear stress values in Fig. 13(a) are so much greater than those in Fig. 13(b).
14. Why is there a strange uplift temperature profile shown in Fig. 14(a) when L=0.01 and rho is very small?
15. The results presented in Fig. 14 are only subjected to very small values of L. To be consistent with rest of the manuscript, you should present those for L = 0, 0.1, 0.2, 0.5, and 1.
Author Response
Dear Reviewer
We would like to thank you for providing a supportive and encouraging
review of the paper. Please find below our response to your
specific points:
“1. The authors mention that these shapes have engineering and practical
value in micromachines. Can you provide some specific examples?”
Thank you for the comment. Indeed, in miniaturized microprocessor
chips, high heat flux should be handled by designing proper
passages of coolant flows, so an efficient heat transfer is necessary by
providing such pipes in computer microchannels equipped with hydrophobic/
slippery surfaces, please refer to Ref.[34] for more details.
This specific application has been added now, please see page 4 lines
3-6.
“2. What are the assumptions of this study?”
Thank you for the remark. The main assumptions, though not
itemized, spread over the text in Section 2. In short, new symmetric
tubular shapes influenced by Navier’s slip at the wall are sought
allowing fully developed thermal and momentum fields.
“3. The definition of the dimensionless temperature doesn’t seem dimensionally
correct. You should double-check to make sure this parameter is dimensionless.”
Thank you for the comment. It has been corrected now, please
see page 5 last line.
“4. The convective terms of the momentum equation drop off due to one of
the assumption that only axial velocity is non-zero. Therefore, only the effects
of pressure gradient and viscous effect are considered in the study. However,
I don’t see how the pressure gradient in the axial direction can be normalized
to become the right terms. You should provide some explanation showing the
steps. I expect a dimensionless parameter to show up in the equation.”
Thank you for the comment. You are perfectly right that convective
terms drop off due to the hydrothermally developed flow
assumption. The only dimensionless parameter resulting from the
normalization is the aspect ratio, please refer to page 6 lines 7-9.
“5. Once again, the convective terms of the energy equation drop off. Because of the lack of explanation in the manuscript, I don’t understand why the conduction in the fluid is simply related to 4 times of the velocity component.
What effect does the velocity component represent in the equation?”
Thank you for the comment. Velocity represents the conductive
term in uTx, where the constant Tx is dropped off during normalization,
please see page 6 lines 7-9.
“6. The authors claimed that the walls are subjected to constant heat flux.
This contradicted to the boundary condition.”
Thank you for the comment. In the considered pipe flow with
the constant heat flux, the form of temperature in Eq.(3) on page 5
enables us to write down zero normalized temperature at the surface
of the pipe, which does not contradict the temperature boundary
condition on page 6.
“7. Based on conventional definition of Nusselt number, I am unable to
derive equation (7). Do show your complete steps.”
Thank you for the comment. Please see that the definition has
been corrected now to follow the formal definition of Nusselt number.
Please refer to the correction in Eq.(3) on page 5, also to arrive at
the Nusselt number formula in Eq.(8) on page 7.
“8. Θb was used to calculate Nusselt number, but how is Θb calculated?”
Thank you for the comment. Please refer to Eq.(8) on page 7.
“9. What if L is greater than unity? Do provide some cases with L > 1.”
Thank you for the comment. Please observe that even the infinite
limit of L has been considered on page 11 Eq.(16), whose corresponding
pictures are exhibited in figure 8 on page 14.
“10. Why does u(r=0) always equals to 2 when L=0 regardless of the values
of rho? Please refer to Fig. 12 and explain.”
Thank you for the comment. It is actually as a result of the obtained
formula in Eq.(22) on page 16, see also Eq.(20) on page 15.
“11. Would it be reasonable to find that u(r=0) approaches infinity when rho
is very small and L is not equal to zero? Please refer to Fig. 12 and explain. Do
remember that this is a Poiseulle flow subjected to a given pressure gradient.”
Thank you for the comment. A discussion has been added now,
please refer to page 17 paragraph 1.
“12. The definition of wall shear stress is not provided in the manuscript.”
Thank you for the comment. Please see that the normalized wall
shear stress based on pressure gradient and hydraulic diameter is
given by Eq.(23) on page 17.
“13. My intuition suggested me that something peculiar happens in Fig.
13(a). Do explain why there exits a bimodal distribution of wall shear stress.
Also, explain why the average wall shear stress values in Fig. 13(a) are so much
greater than those in Fig. 13(b).”
Thank you for the comment. As suggested from the velocity profiles
trough figures 9-11, profiles with smaller aspect ratio are likely to
produce higher velocity gradients at the azimuths resulting in larger
shears at the tube surfaces as compared to those with bigger aspect
ratio pipes. Moreover, the existence of bimodal distribution of wall
shear stress for tinier pipes can be attributed to higher velocities
prevalent to such shapes. These explanations have been added to
text now, please refer to page 18 last 3 lines and page 19 first 2 lines.
“14. Why is there a strange uplift temperature profile shown in Fig. 14(a)
when L=0.01 and rho is very small?”
Thank you for the comment. It was due to the code (insufficient
precision), which has been improved now, please see page 20 Figure
14(a).
“15. The results presented in Fig. 14 are only subjected to very small values
of L. To be consistent with rest of the manuscript, you should present those for
L = 0, 0.1, 0.2, 0.5, and 1.”
Thank you for the comment. The reason deliberately we pick
smaller values of L for temperature solutions is that they are produced
from a small perturbation L analysis, only up to order of L, please
refer to pages 8 and 9. Higher values of L, therefore, are avoided
not to present misleading characters, otherwise they must be justified
from a full numerical simulation. This has been explained now, please
see page 21 lines 1-4.
Round 2
Reviewer 1 Report
The authors addressed all comments and the manuscript could be published in the journal.
Author Response
Thank you for accepting our work after revision.
Reviewer 2 Report
1. The assumptions of this study should be clearly addressed. Clearly, these assumptions were not complete.
2. The derivation for the dimensionless momentum and energy equations are still unclear. The rebuttal does not answer my previous questions.
3. The authors claimed that the walls are subjected to constant heat flux.
This contradicted to the boundary condition. Even if the authors have redifined the dimensionless temperature, it is still WRONG!
4. The approach to calculate the Nusselt number formula in Eq.(8) on page 7 is also WRONG.
5. Why does u(r=0) always equals to 2 when L=0 regardless of the values
of rho? Please refer to Fig. 12 and explain. Do answer the based on physics (or fluid mechanics points of view) NOT based on the mathematical equation. What if your solution is wrong?
6. The normalized wall shear stress based on pressure gradient and hydraulic diameter is given by Eq.(23) on page 17. Apparently, tau is calculated based on radial and tangential velocity components. However, these two components are zero as you mentioned in the manuscript. If this is how you calculate your shear stress, then it should be zero.
7. It is said that "the existence of bimodal distribution of wall shear stress for tinier pipes can be attributed to higher velocities prevalent to such shapes." Please seriously explain the phenomena. Your statement does not stand on a solid groud.
8. I am finally glad that you have pointed out that L is your perturbed term. If so, would it be meaningful when you consider L larger than 0.1? In Fig. 8, L even approaches infinity. For your information, perturbation approach is a powerful tool but MUST be handled with carefulness. You should have checked the convergencce of your series solution to identify the limitation of your series solution.
Round 3
Reviewer 2 Report
First of all, I would like to emphasize that I have no bias against your work. I am just fulfilling my duty as a reviewer to make sure that a technical publication should not be sending out wrong technical information to our readers. Also, as a responsible scholar, it is our responsibility to take publication very seriuosly. I do respect you two as mathematicians. However, many mathematicians do formulating a wrong fluid mathematical problem and therefore presenting a "fluid" solution that does not make sense in the eyes of fluid scholars.
Therefore, I wish to see reasonable insight why a fluid velocity always equals to 2 (Do refer to my question #5 in my previous comment). It is not our duty as a reviewer to verify your answer and certainly not to prove you are wrong. I did not say that your answer was wrong. I said "what if". Making physical interpretation based on mathematical equation should be supported by phyical meaning.
To make my point clear, I just want to raise two thoughts.
You have provided a paper published by Shahmadan et. al. This is really great to help readers follow how to come out with the equations. On page 133, it said, "Inserting Eqs. (2) and (7) into Eq. (1c) and considering the fully developed rectilinear flow (setting the transverse velocity components to zero), the following dimensionless form of the heat transfer equation is obtained:"
If you follow exactly the instruction, i.e., inserting Eqs. (2) and (7) into Eq. (1c) and setting the transverse velocity components to zero, the heat transfer equation (1c) will not become equation (8). Maybe I am too old or too careless. I wish you can enlighten me, prove me wrong, and show me how equation (8) is possible in the form presented in the paper. In physics, equation (8) literally means, the heat conduction is controlled by 4 times of the fluid velocity. I could never understand this.
Your temperature was normalized using q, which represents the constant heat flux at the wall (eq. 3 in your manuscript). According to your wall boundary condition (eq. 5), THETA = 0. This means you are solving a problem that subjects to constant wall temperature. On one hand, you are solving a constant wall temperature problem, but on the other hand, you are implicitly assuming there exist a constant heat flux at wall. This is only possible if the heat transfer mechanism in the microtube is solely based on heat conduction. However, your velocity is at least in the order of the bulk velocity in the microtube (please see figures 11), implying that the effect of heat convection should not be ignored. Then, the physics contradict by itself. Perhaps, you can show me the points where I have not seen correctly or misunderstand your approach.
Once again, I do respect your profession and expertise in mathamatics. What I would like to do is to make sure you bring in reasonable physics to my fluid community. I have to admit mathematic approaches have helped the development of many fluid theories and improved our understand of fluid nature.
Round 4
Reviewer 2 Report
Tx = qp / (rho * cp * A * ub) is ONLY correct for mean temperature. Clearly, the variable T you solved in your manuscript is NOT the mean temperature. I don't think your work has based on a consistant assumption.
The concept in Ali2007 (page 22) is correct. The wall temperature has never been assumed constant. He did not normalize the temperature. However, you non-dimensionalized your temperature. These are two different scenario.
I think I have wasted enough time trying to show your inconsistency. I have a feeling that all you care is to publish regardless of a scientist's persistence for clarification and truth. I don't see the need of my further contribution. Good luck with your publication.